# Social support, social context and nonadherence to treatment in young senior patients with multimorbidity and polypharmacy followed-up in primary care. MULTIPAP Study

**Cristina M. Lozano-Hernández**[1,2,3,4]*, **Juan A. López-Rodríguez**[1,3,4,5,6], **Francisca Leiva-Fernández**[3,7,8], **Amaia Calderón-Larrañaga**[3,9,10,11], **Jaime Barrio-Cortes**[1,4], **Luis A. Gimeno-Feliu**[3,11,12,13], **Beatriz Poblador-Plou**[3,11,14], **Isabel del Cura-González**[1,3,5], **MULTIPAP GROUP**¶

1 Research Unit, Primary Health Care Management, Madrid, Spain, 2 Interuniversity Doctoral Program in Epidemiology and Public Health, Rey Juan Carlos University, Alcorcon, Madrid, Spain, 3 Research Network in Health Services in Chronic Diseases (REDISSEC) ISCIII, Madrid, Spain, 4 Biosanitary Research and Innovation Foundation of Primary Care (FIIBAP), Madrid, Spain, 5 Department of Medical Specialties and Public Health, Faculty of Health Sciences, Rey Juan Carlos University, Madrid, Spain, 6 General Ricardos Primary Health Care Centre, Madrid, Spain, 7 Multiprofessional Teaching Unit for Family and Community Care Primary Care District Málaga-Guadarhorce, Málaga, Spain, 8 Biomedical Research Institute of Malaga-IBIMA, Andalusian Health Service, Málaga, Spain, 9 Joint Action on Chronic Diseases (JA-CHRODIS) European Commission, Brussels, Belgium, 10 Aging Research Centre, Department of Neurobiology, Care Sciences and Society, Karolinska Institute & Stockholm University, Stockholm, Sweden, 11 EpiChron Research Group on Chronic Diseases, Aragonese Institute of Health Sciences (IACS), IIS Aragón, Zaragoza, Spain, 12 San Pablo Primary Health Care Centre, Aragon Health Service, Zaragoza, Spain, 13 Department of Medicine, Psychiatry and Dermatology, University of Zaragoza, Zaragoza, Spain, 14 Miguel Servet University Hospital, Zaragoza, Spain

¶ Membership of the MULTIPAP GROUP is provided in the Acknowledgments.
* cristinamaria.lozano@salud.madrid.org

## Abstract

### Objective

To estimate the prevalence of nonadherence to treatment and its relationship with social support and social context in patients with multimorbidity and polypharmacy followed-up in primary care.

### Methods

This was an observational, descriptive, cross-sectional, multicenter study with an analytical approach. A total of 593 patients between 65–74 years of age with multimorbidity (≥3 diseases) and polypharmacy (≥5 drugs) during the last three months and agreed to participate in the MULTIPAP Study. The main variable was adherence (Morisky-Green). The predictors were social support (structural support and functional support (DUFSS)); sociodemographic variables; indicators of urban objective vulnerability; health-related quality of life (EQ-5D-5L-VAS & QALY); and clinical variables. Descriptive, bivariate and multivariate analyses with logistic regression models and robust estimators were performed.

**Data Availability Statement:** Regarding data exchange, the Aragon Ethics Committee approved this research without considering the option of data sharing. The data contains sensitive clinical information about the patient, so there are ethical and legal restrictions to sharing the data set. The data are part of the MULTIPAP study and can be requested by contacting the Aragon Ethics Committee at the email address ceica@aragon.es; for the request of data you can also contact the Primary Care Management of Madrid at the email address gap@salud.madrid.org; and by contacting the Technical Direction of Teaching and Research at the email address dtdei@salud.madrid.org. The MULTIPAP Group may establish future collaborations with other groups based on the same data. The main researchers of the project will be contacted (Alexandra Prados-Torres at sprados. iacs@aragon.es; Daniel Prados-Torres at uand. prados.sspa@juntadeandalucia.es; and Isabel del Cura at isabel.cura@salud.madrid.org). However, each new project based on these data must be previously submitted to CEICA for approval.

**Funding:** This study was funded by the Fondo de Investigaciones Sanitarias ISCIII (Grant Numbers PI15/00276, PI15/00572, PI15/00996), REDISSEC (Project Numbers RD16/0001/0006, RD16/0001/ 0005 and RD16/0001/0004), and the European Regional Development Fund ("A way to build Europe"). Funders had no role in study design or in the decision to submit the report for publication.

**Competing interests:** The authors have declared that no competing interests exist.

## Results

Four out of ten patients were nonadherent, 47% had not completed primary education, 28.7% had an income ≤1050 €/month, 35% reported four or more IUVs, and the average perceived health-related quality of life (HRQOL) EQ-5D-5L-VAS was 65.5. The items that measure functional support, with significantly different means between nonadherent and adherent patients were receiving love and affection (-0.23; 95%CI: -0.40;-0.06), help when ill (-0.25; 95%CI: -0.42;-0.08), useful advice (-0.20; 95%CI: -0.37;-0.02), social invitations (-0.22; 95%CI:-0.44;-0.01), and recognition (-0.29; 95%CI:-0.50;-0.08). Factors associated with nonadherence were belonging to the medium vs. low tertile of functional support (0.62; 95%CI: 0.42;0.94), reporting less than four IUVs (0.69; 95%CI: 0.46;1.02) and higher HRQOL perception (0.98; 95%CI: 0.98;0.99).

## Conclusions

Among patients 65–74 years of age with multimorbidity and polypharmacy, lower functional support was related to nonadherence to treatment. The nonadherence decreased in those patients with higher functional support, lower urban vulnerability and higher perceived health status according to the visual analog scale of health-related quality of life.

## Introduction

Population aging has led multimorbidity to be becoming increasingly prevalent in the adult population of most Western European countries [1]. Most studies define multimorbidity as the concurrent presence of two or more, or three or more chronic diseases; the latter definition is more suitable for the identification of patients with complex health needs [2]. It is estimated that the average number of chronic conditions in those over 75 years is 3.2, while that for the young seniors (65–74 years) is 2.8 [1–3]. Multimorbidity affects 81.5% of people older than 85 years, 62% of those 65–74 years old and 50% of those younger than 65 years old [4]. Multimorbidity is associated with polypharmacy, defined as the simultaneous consumption of five or more drugs [5]. Polypharmacy has undesirable consequences, such as increased risk of potentially inappropriate medication, misuse of doses, either by excess or by default of necessary treatments, nonadherence and increased risk of interactions and adverse drug reactions [6].

Studies on different population groups with respect to age and chronic condition showed an average of 50% adherence to long-term therapy. They found that about half of the chronic patients do not comply with their prescription or do so incorrectly, particularly with respect to time, dose, frequency and duration [7,8]. Nonadherence to prescribed treatment is a growing and complex problem for both patients and healthcare systems [9,10]. Haynes et al. [11] identified more than 250 factors that can influence drug adherence, among which being older and social isolation stood out as the most important ones [12]. Components of the social context are all those that encompass the individual's living conditions: the physical environment in which they live, the socio-economic level, the level of education, work, income level, the social network and social support [13]. The challenge of improving adherence has traditionally focused on the individual characteristics of the patient, the complexity of the treatment, the type of information provided, health literacy, and the physician-patient relationship [14,15], with social factors being little explored [8,16]. Thus far, some studies have found a significant relationship between adherence and the social context, in terms of socioeconomic status [17]

and social support [18,19]. However, there are few studies exploring the impact of all these determinants together, which hinders the comprehensive approach necessary to study nonadherence to treatment [20].

The definition of social support includes whether the basic social needs of a person (affection, esteem, approval, belonging, identity and safety) are satisfied through the interaction with others [21]. It covers three areas: (1) structural, which assesses the number and pattern of direct and indirect bonds surrounding the individual; (2) functional, which corresponds to the different types of resources that flow through the bonds of the social network; and (3) informative, which reflects the knowledge provided to the individual through his or her social network [22,23]. Two meta-analyses have analyzed the relationship between social support and adherence to treatment, concluding that adherence improves more significantly with functional support than with structural support [20,24]. Having someone to talk to and have contact with, as well as someone who in turn provides emotional support, seem to play a prominent role in the association of functional support and adherence to treatment [18]. Moreover, the source of functional support and the type of help that is received also have a considerable influence on this association [19,24,25]. In this sense, jointly studying the different elements of the social context and exhaustively exploring the influence of functional support on nonadherence could be key to better understand this complex framework and the role that each factor plays in therapeutic adherence [19]. The main objective of this study is to estimate the prevalence of nonadherence to treatment and its relationship with social support and social context in patients with multimorbidity and polypharmacy followed-up in primary care.

## Materials and methods

An observational, descriptive, cross-sectional study with an analytical approach was performed using baseline data from the MULTIPAP Study [26]. This study was a pragmatic group-controlled, randomized clinical trial with 12 months of follow-up conducted in 38 health centers in the regions of Andalucía, Aragon and Madrid (Spain) with the participation of 117 family physicians (GPs), each recruiting five patients. Patients aged 65–74 years with multimorbidity (≥3 chronic diseases) and polypharmacy (≥5 different drugs for at least the last three months) who visited their GP at least once in the last year and had given their written informed consent to participate in the MULTIPAP Study were included [26]. We excluded institutionalized patients with severe mental illness and those with a life expectancy of less than 12 months according to their physician. Patients were selected by random sampling among those who met the inclusion criteria. All the variables described below were collected by the GP through an interview during the consultation. The participating GPs were previously trained to conduct the interview through an electronic data collection notebook.

The main outcome variable was adherence to treatment (adherent/nonadherent), measured with the Morisky-Green test. The Morisky-Green questionnaire asks a series of closed questions to the patient: "*do you ever forget to take the medicines to treat your illness; do you take the medicines at the indicated times; when you feel well, do you stop taking the medicine; if you ever feel bad, do you stop taking the medicine? Patients are considered adherent if they answer all four questions correctly and non-adherent if they answer three or fewer questions*" [27]. The main independent variable was social support, measured in two ways: structural support (information about marital status and number of cohabitants in the home) and functional support. The latter was measured through the Duke UNC-11 Functional Social Support (DUFSS). This questionnaire offers a total score of functional support and two additional scores related to each of the domains revealed by its factor analysis: confidential and affective support [28]. The version used in this study was composed of 11 items. Each item uses a Likert-type

response scale from 1 ("Much less than I would like") to 5 ("As much as I would like") [29]. Since its creation to date, the instrument has been validated in very different populations, showing differences in the distribution of the items that make up each of the domains. Validation performed by Ayala et al. [30] was carried out in a noninstitutionalized Spanish population, with an average age of 72 years. As this is a population similar to ours, we have used the result of their factor analysis, in which confidential support is measured through seven items (4, 5, 6, 7, 8, 10 and 11), with a total score of 35, and affective support is measured through 4 items (1, 2, 3, and 9), with a total score of 20. The total score for social support was categorized in tertiles, being tertile 1 the lowest.

The following potential predictor variables were considered: a) sociodemographic variables such as age, sex, retirement status (retirement/no retirement), social class (collected through the 7 categories of the CNO-SEE12 instrument and subsequently grouped) [31], education level (primary education incomplete, primary education, secondary or higher education), and socioeconomic level (≤1050€/month, 1051–2250€/month and ≥2251€/month); b) indicators of subjective urban vulnerability (IUVs), based on those collected by the National Health Survey to explore participants' neighborhood (noise level, odors, poor-quality drinking water, unclean streets, air pollution, lack of green areas, feral animals and crime) [32]; and c) clinical factors (number of chronic diseases and number of drugs consumed) and health-related quality of life (HRQOL) measured by the EQ-5D-5L [33]. The EQ-5D-5L determines the perceived general state of health measured by a visual analog scale (VAS) and utilities.

A descriptive analysis of the characteristics of the patients was performed, with frequencies and percentages for the qualitative variables and with means and standard deviations (SD) or medians and interquartile ranges (IQR) for the quantitative variables according to their distribution. The prevalence of nonadherence was estimated with the 95% confidence intervals (CI). The contrast of qualitative variables was performed with the Pearson Chi-squared test, and the contrast of normally distributed quantitative variables was performed with Student's t-test. To study the association between functional support (independent variable) and nonadherence (dependent variable), a logistic regression model was fit with sequential forward fitting in three steps. Model 1 was adjusted for sociodemographic factors (age, sex, retirement status, social class, education level and socioeconomic status). Model 2 was adjusted additionally for IUVs (<4 vs. ≥4 indicators). A final model was constructed (model 3) adjusting additionally for clinical factors (number of diseases and number of drugs) and health-related quality of life. Considering that the patients were included in the study by cluster-sampling (each GP included five patients), robust estimators were obtained. The data analysis was performed with the statistical software STATA v14.

## Ethical and legal aspects

The project was approved by the Clinical Research Ethics Committee of Aragon (CEICA) on September 30, 2015, with the reference number PI15/0217. And has been favorably evaluated by the Research Ethics Committee of the Province of Malaga on September 25, 2015, and by the Central Committee of Primary Care Research of the Community of Madrid.

## Results

Of the 593 patients included in the study, a total of 40.8% (95% CI: 36.9%; -44.8%) were nonadherent. Table 1 shows the distribution of variables and the number of subjects (total population, nonadherent and adherent). More than half of the sample (56.3%) were women, and the mean age was 69.7 years (SD 2.7 years). Compared to adherent patients, those who were nonadherent reported a higher percentage of 4 or more IUVs (48.3% vs. 36.8%, p = 0.007) and had

**Table 1. Characteristics of patients according to adherence.**

| | Total n (%) | Adherent n (%) | Nonadherent n (%) | p-value |
|---|---|---|---|---|
| **N** | 593(100) | 351(59.2) | 242(40.8) | |
| Age * | 69.7(2,7) | 69.8(2.7) | 69.6(2.7) | 0.55 |
| **Sex** | | | | |
| Male | 259(43.7) | 153(59.1) | 106(40.9) | 0.96 |
| Female | 334 (56.3) | 198(59.3) | 136(40.7) | |
| **Retirement status** | | | | |
| Retired | 538(90.7) | 321(59.7) | 217(40.3) | 0.46 |
| Nonretired | 55(9.3) | 30(54.6) | 25(45.5) | |
| **Social class** | | | | |
| Mid-level supervisors and directors | 234(39.5) | 147(62.8) | 87(37.2) | 0.33 |
| Skilled primary sector | 217(36.6) | 125(57.6) | 92(42.4) | |
| Unskilled | 142(24) | 79(55.6) | 63(44.4) | |
| **Education level** | 74(12.5) | 43(58.1) | 31(41.9) | |
| Primary education incomplete | 279(47.1) | 163(58.4) | 116(41.6) | 0.88 |
| Primary education | 240(40.5) | 145(60.4) | 95(39.6) | |
| Secondary or higher education | | | | |
| **Socioeconomic level** | | | | |
| ≥2251€/month | 59(10) | 37(62.7) | 22(37.3) | |
| 1051–2250€/month | 342(57.7) | 199(58.2) | 143(41.8) | 0.45 |
| ≤1050€/month | 170(28.7) | 105(61.8) | 65(38.2) | |
| NS/NC | 22(3.7) | 10(45.5) | 12(54.6) | |
| **Urban vulnerability indicators** | | | | |
| <4 indicators | 386(65.1) | 244(63.2) | 142(36.8) | **<0.01** |
| ≥4 indicators | 207(34.9) | 107(51.7) | 100(48.3) | |
| **Number of diseases**** | 6(4–7) | 6(4–7) | 5.5(4–7) | 0.99 |
| **Number of diseases*** | 6.1(2.5) | 6.1(2.5) | 6.1(2.4) | 1.00 |
| **Number of drugs**** | 7(6–9) | 7(5–9) | 7(6–9) | 0.92 |
| **Number of drugs*** | 7.4(2.4) | 7.5(2.5) | 7.4(2.2) | 0.57 |
| **HRQOL *** | | | | |
| EQ-5D-5L-VAS | 65.5(20.5) | 68.3(19.7) | 61.5(21.1) | **<0.01** |
| Utilities | 0.77(0.2) | 0.79(0.2) | 0.75(0.2) | **0.02** |
| **Structural social support** | | | | |
| **Cohabitation** | | | | |
| Lives alone | 106(17.9) | 59(55.7) | 47(44.3) | 0.72 |
| Lives with 1 person | 368(62.1) | 221(60) | 147(40) | |
| Lives with ≥2 people | 119(20.1) | 71(59.7) | 48(40.3) | |
| **Marital status** | | | | |
| Single | 23(3.9) | 11(47.8) | 12(52.2) | 0.72 |
| Married or partner | 447(75.4) | 268(60) | 179(40) | |
| Separated | 29(4.9) | 17(58.6) | 12(41.4) | |
| Widow | 94(15.8) | 55(58.5) | 39(41.5) | |
| **Functional social support (DUFSS)** | | | | |
| Total score* | 43.7(8.8) | 44.5(8.1) | 42.7(9.6) | **0.01** |
| 1st tertile (low) | 190(32) | 97(51.1) | 93(49) | |
| 2nd tertile (medium) | 191(32.2) | 123(64.4) | 68(35.6) | **0.01** |
| 3rd tertile (high) | 212(35.8) | 131(61.8) | 81(38.2) | |
| **Confidential score*** | 29.5(5.9) | 30(5.5) | 28.8(6.4) | **0.03** |

*(Continued)*

**Table 1.** (Continued)

| | Total n (%) | Adherent n (%) | Nonadherent n (%) | p-value |
|---|---|---|---|---|
| **Affective score**[*] | 14.2(3.7) | 14.5(3.4) | 13.8(4) | **0.02** |

[*]mean (SD)

[**]median (IQR)

a lower EQ-5D-5L-VAS score (68.3 vs. 61.5, p < 0.01) and utilities (0.79 vs. 0.75, p = 0.02). There weren't any statistically difference between adherent and nonadherent patients regard to number of diseases and drugs in our results.

Regarding structural support, neither number of cohabitants nor marital status were associated to treatment adherence. The mean score for functional support was 43.7 (SD 8.8) out of 55 points; for the confidential domain, 29.5 (SD 5.9) out of 35 points, and for the emotional domain, 14.2 (SD 3.7) out of 20 points. A statistically significant association was found between functional support and adherence, both for the total score and its domains (Table 1).

The detailed study of functional support showed significantly different means between adherent and nonadherent patients for the following items: receiving love and affection (-0.23; 95% CI: -0.40; -0.06), receiving help when sick in bed (-0.25; 95% CI: -0.42;-0.08), receiving useful advice about important events (-0.20; 95% CI: -0.37;-0.02), receiving social invitations (-0.22; 95% CI: -0.44;-0.01), and receiving praise and recognition at work (-0.29; 95% CI: -0.50;-0.08) (Table 2).

In the adjusted models, nonadherence was associated with having a medium vs. high functional support (OR 0.62; 95% CI 0.41–0.94), less than 4 IUVs (OR 0.66; 95% CI 0.44–1.99) and higher EQ-5D-5L-VAS score (OR 0.98; 95% CI 0.98–0.99) (Tables 3 & 4).

## Discussion

### Main findings of the study

In patients aged 65–74 years with multimorbidity and polypharmacy, nonadherence to treatment was moderate. Lower functional support was associated with treatment nonadherence in these patients, although this effect was not consistent across all levels of support. The specific

**Table 2. Distribution of the DUFSS questionnaire items according to adherence.** Order of items following Ayala´s proposal.

| | Total | Adherent | Nonadherent | Difference of means | p-value |
|---|---|---|---|---|---|
| Confidential Domain Mean (SD) | | | | | |
| **Item 7.** I have chances to talk to someone I trust about my personal and family problems | 4.2(1.1) | 4.1(1.1) | 4.2(1.1) | -0.05 (-0.24–0.13) | 0.56 |
| **Item 8.** I have chances to talk to someone about money problems | 4.1(1.1) | 4(1.2) | 4.1(1.1) | -0.11(-0.29–0.08) | 0.25 |
| **Item 6.** I have chances to talk to someone about problems at work or at home | 4.2(1.1) | 4.1(1.2) | 4.2(1.1) | -0.13(-0.31–0.05) | 0.17 |
| **Item 5.** I receive love and affection | 4.3(1) | 4.2(1.2) | 4.4(0.9) | -0.23(-0.40-(-0.06)) | **<0.01** |
| **Item 4.** I have people who care what happens to me | 4.4(1) | 4.3(1.1) | 4.4(0.9) | -0.12(-0.28–0.05) | 0.16 |
| **Item 11.** I receive help when I am sick in bed | 4.3(1.1) | 4.2(1.2) | 4.4(0.9) | -0.25(-0.42-(-0.08)) | **<0.01** |
| **Item 10.** I receive useful advice about important things in my life | 4(1.1) | 3.9(1.1) | 4.1(1) | -0.20(-0.37-(-0.02)) | **0.03** |
| Affective Domain Mean (SD) | | | | | |
| **Item 2.** I receive help in matters related to my home | 3.1(1.4) | 3.1(1.4) | 3.1(1.3) | -0.05(-0.27–0.33) | 0.69 |
| **Item 1.** I get visits from friends and family | 3.6(1.3) | 3.5(1.3) | 3.7(1.3) | - 0.17(-0.38–0.04) | 0.12 |
| **Item 9.** I receive invitations to participate in activities and go out with other people | 3.8(1.3) | 3.7(1.3) | 3.9(1.2) | -0.22(-0.44-(-0.01)) | **0.04** |
| **Item 3.** I receive praise and recognition when I do my job well | 3.6(1.3) | 3.5(1.3) | 3.8(1.2) | -0.29(-0.50-(-0.08)) | **<0.01** |

**Table 3. Factors associated with nonadherence to treatment in patients with multimorbidity and polypharmacy.**

| | Model 0 | | Model 1 | | Model 2 | | Model 3 | |
|---|---|---|---|---|---|---|---|---|
| | OR (CI 95%) | p-value | OR (CI 95%) | p-value | OR (CI 95%) | p-value | OR (CI 95%) | p-value |
| Functional support 1st tertile (low) | ref | | ref | | ref | | ref | |
| 2nd tertile (medium) | 0.58(0.38–0.87) | <**0.01**0 | 0.59(0.39–0.89) | **0.01** | 0.61(0.40–0.92) | **0.02** | 0.62(0.41–0.94) | **0.03** |
| 3rd tertile (high) | 0.64(0.41–1.02) | .06 | 0.65(0.41–1.04) | 0.07 | 0.71(0.45–1.12) | 0.14 | 0.72(0.45–1.14) | 0.16 |
| **Age** | --- | | 1.00(0.93–1.06) | 0.75 | 0.99(0.93–1.06) | 0.87 | 1.00(0.93–1.06) | 0.92 |
| **Sex** | --- | | | | | | | |
| Female | | | ref | | ref | | ref | |
| Male | | | 1.07(0.75–1.53) | 0.71 | 1.08(0.76–1.54) | 0.68 | 1.19(0.83–1.71) | 0.34 |
| **Retirement status** | | | | | | | | |
| Retired | --- | | ref | ref | ref | ref | ref | |
| Nonretired | | | 1.19(0.59–2.38) | 0.62 | 1.23(0.61–2.47) | 0.56 | 1.28(0.64–2.57) | 0.49 |
| **Social class** | --- | | | | | | | |
| Unskilled | | | ref | | ref | | ref | |
| Skilled primary sector | | | 0.92(0.61–1.38) | 0.69 | 0.92(0.61–1.38) | 0.68 | 0.96(0.69–1.46) | 0.84 |
| Mid-level supervisors and directors | | | 0.72(0.44–1.18) | 0.20 | 0.71(0.43–1.18) | 0.18 | 0.77(0.46–1.29) | 0.31 |
| **Education level** | --- | | | | | | | |
| Primary incomplete | | | ref | | ref | | ref | |
| Primary education | | | 0.99(0.66–1.47) | 0.95 | 0.98(0.66–1.45) | 0.91 | 1.03(0.69–1.53) | 0.90 |
| Secondary or higher education | | | 1.20(0.66–2.19) | 0.55 | 1.24(0.68–2.29) | 0.49 | 1.27(0.68–2.38) | 0.44 |
| **Socioeconomic level** | --- | | | | | | | |
| ≤1050€/month | | | ref | | ref | | ref | |
| 1051–2250€/month | | | 1.26(0.84–1.87) | 0.26 | 1.24(0.83–1.84) | 0.30 | 1.18(0.79–1.77) | 0.41 |
| ≥2251€/month | | | 1.12(0.58–2.13) | 0.74 | 1.10(0.57–2.11) | 0.77 | 1.08(0.56–2.09) | 0.82 |
| NS/NC | | | 1.96(0.77–4.98) | 0.16 | 2.12(0.83–5.45) | 0.12 | 1.80(0.73–4.44) | 0.20 |
| **Urban vulnerability indicators** | --- | | --- | | | | | |
| ≥4 indicators | | | | | ref | **0.03** | ref | **0.05** |
| <4 indicators | | | | | 0.62(0.41–0.94) | | 0.66(0.44–0.99) | |
| **Number of diseases** | --- | | --- | | --- | | 0.99(0.92–1.07) | 0.85 |
| **Number of drugs** | --- | | --- | | --- | | 0.95(0.88–1.03) | 0.22 |
| **EQ-5D-5L-VAS** | --- | | --- | | --- | | 0.98(0.97–0.99) | <**0.01** |
| **Utilities** | --- | | --- | | --- | | 1.10(0.36–3.38) | 0.87 |
| Pseudo $R^2$ | 0.0099 | | 0.0168 | | 0.0253 | | 0.0418 | |

**Table 4. Factors associated with nonadherence to treatment in patients with multimorbidity and polypharmacy.**
Final model with those variables that were significantly associated in Table 3.

| | OR (CI 95%) | p-value |
|---|---|---|
| **Functional support** | | |
| 1st tertile (low) | ref | |
| 2nd tertile (medium) | 0.62(0.42–0.94) | 0.023 |
| 3rd tertile (high) | 0.73(0.47–1.14) | 0.165 |
| **EQ-5D-5L-VAS** | 0.98(0.98–0.99) | <**0.01** |
| **Urban vulnerability indicators** | | |
| ≥4 indicators | ref | |
| <4 indicators | 0.69(0.46–1.02) | 0.061 |
| **Number of drugs** | 0.94(0.88–1.01) | 0.085 |
| Pseudo $R^2$ 0.0354 | | |

items associated with nonadherence were those related to affection, help in the disease, advice, social invitations and recognition. Nonadherence to treatment was associated with greater urban vulnerability and lower health-related quality of life.

## Strengths and comparison with other studies

Among the strengths of our study, it is worth mentioning, on the one hand, the selected population, representative of the general population with these characteristics and hardly studied in previous research. The older youth with multi-morbidity and polypharmacy is a very frequent population among the adult population in most Western European countries, with a great potential for action in terms of optimizing adherence to treatment. On the other hand, the study of socioeconomic factors with nonadherence to treatment -with special emphasis on functional support-. Furthermore, the models were adjusted by a large number of clinical, sociodemographic and urban vulnerability variables, which favors a better understanding of the mechanisms underlying the associations found.

The nonadherence to treatment in our study is somewhat lower than that obtained in the meta-analysis conducted by Naderi et al. (43% in patients with an average age of 64 years), and another study by Alves et al. in which a prevalence of 50% was found in a population with multimorbidity and with a mean age of 56.5 years [34,35]. These differences can be explained by the age ranges included in each of the studies, where, as reported by Feehan et al., younger age is associated with lower levels of adherence [36].

In relation to the influence of functional support on treatment adherence, it is difficult to compare our results with those obtained in other studies because of differences in study populations, questionnaires and theoretical frameworks. A study conducted in the field of psychology concluded that there is an inverse association between social support and nonadherence, without obtaining a statistical significance. However, it excluded people over 65 years of age with multiple chronic diseases [7]. In a meta-analysis by Dimatteo et al., they found no statistical significance in the association between structural support and nonadherence to treatment, although they did find a significant association for functional support [19], as was the case in our study. A study by Mondesir et al., conducted in a population of chronic patients with a mean age of 66.2 years, found that having care support while having a disease or disability increased adherence to treatment [25], which coincides with the results obtained for item 11 of the DUFSS scale in our study.

An interpretative qualitative meta-analysis that had, among others, the objective of exploring the experiences lived by patients in the context of multimorbidity indicated that patients who have structural support may feel isolated if they believe that people in their environment do not want to understand their problems [37]. Additionally, another qualitative study on patients with chronic obstructive pulmonary disease concluded that this sense of incomprehension from their environment can lead people to passively cope with their disease [38]. Items that relate to the need to speak or receive support for aspects beyond the disease were not significantly associated with nonadherence in our study, but this may be due to the social needs and lifelong characteristics of the people who make up the sample. In young seniors individuals, their needs for social support could be more related to maintaining their affective, social and recognition relationships, requiring explicit help only for important events and situations of illness.

In the present study, indicators of subjective urban vulnerability were related to nonadherence. Two recent qualitative studies concluded that neighborhoods affect the health of residents by creating a social context that directly influences their beliefs and behaviors, such as adherence to prescribed treatment [39,40]. The fact that in our study nonadherence was associated with urban vulnerability and not with social class or income level could be because

people in this age range have more homogenous salaries coming from their pensions despite living in communities with varying social statuses. The urban vulnerability indicators may capture such heterogeneity in the socioeconomic and social class of the study patients throughout their working life. However, more studies are needed to delve into this relationship.

Regarding the state of perceived health according to the health-related quality of life VAS, the score obtained in our study is similar to that of another study performed in our country with people over 65 years of age who were not institutionalized, for whom a mean score of 66.6 (95% CI 65.3–68) was observed [41]. A worse perceived health-related quality of life has been previously associated with nonadherence to treatment. Alves et al. [35] found that nonadherence to treatment was associated with a poorer health-related quality of life in patients with chronic kidney disease. On the other hand, Mclane C. et al. found that patients with a better health-related quality of life had higher nonadherence rates, given that their health problem did not affect their daily routine [42].

Concerning the number of diseases and the number of drugs, which in other studies have been associated with adherence to treatment [39,43], these were not statistically significant in our case. This could be due to the fact that ours is a very homogeneous sample with respect to the clinical variables collected, since all of the patients presented multimorbidity and polypharmacy.

## Limitations

Methods for measuring adherence to treatment based on self-reported information may have certain limitations due to recall bias, social desirability and faults in self- observation [44]. However, the Morisky-Green test has been validated and has high specificity, a high positive predictive value and is easy to perform [27]; therefore, it is widely used.

In relation to social support, there are different definitions and measurement instruments depending on the study discipline. Even studies carried out in the field of health that use the DUFSS questionnaire to measure functional support present important differences concerning the items included in the different domains and the reporting of results. This questionnaire has been validated in different population groups [45–49]. Among the Spanish validations, three of the studies were conducted in primary care. Among them, De la Revilla et al. and Bellón et al. validated it for socioeconomically disadvantaged women in the general population [28,50]. Cuellar-Flores et al. and Mas-Expósito et al. validated the instrument in caregivers and mental health patients [51,52]. In our study, we chose to use the most recent validation performed by Ayala et al. [30] in people 60 years or older who were not institutionalized given the similarities with our population. As for the interpretation of the results, there is also variability because authors can report their results either qualitative or quantitatively. In this study, we did not limit our exposure to the level of support, taking into account the subjectivity and variability of this interpretation depending on social, economic and cultural contexts [53].

Adherence is influenced by a large number of factors. Haynes estimated up to 250 factors [11], many of which have not been included in our study. This may explain why the relationship between nonadherence and social support does not hold up for higher levels of support.

The study of the social context implies the difficulty of studying strongly related multidimensional concepts. In order to study the social context, the present work collects the educational level, social class, educational level, physical environment through the indicators of subjective urban vulnerability and the monthly income of the family unit.

## Implications of the study findings

By knowing the role that social support plays in relation to health behaviors and adherence, health professionals can anticipate a more effective approach, taking into account the

socioeconomic context of the patient, their health-related quality of life and preferences. The results of this study reveal that therapeutic nonadherence is not related to the quantification of the structural support of the person but, rather, to the importance that people confer to the support they receive and when they receive it. The perception of these patients concerning the environment in which they live and their health-related quality of life seems to explain this association too. Both urban vulnerability and health-related quality of life are complex and multidimensional concepts conditioned by the subjective perception of the person, reflecting aspects linked to both the physical and social environment, as well as individual-level life and health experiences. It is therefore essential to integrate these patient-reported outcomes in clinical practice to improve the care of older people with multimorbidity, enabling also to individualize their treatments and increase their involvement in self-care.

## Acknowledgments

To our colleagues from the Research Unit Primary Health Care Management Madrid for their support. To all the professionals from the participant Primary Healthcare Centers. To all patients for their contribution to this research.

## MULTIPAP GROUP

**Lead authors for the MULTIPAP Study group:** Alexandra Prados Torres (Aragonese Institute of Health Sciences (IACS), IIS Aragón, Miguel Servet University Hospital, Spain) sprados. iacs@aragon.es, Juan Daniel Prados Torres (Multiprofessional Teaching Unit for Family and Community Care Primary Care District Málaga-Guadarhorce. Málaga) juand.prados.sspa@-juntadeandalucia.es, Isabel del Cura (Research unit. Primary Health Care Management Madrid. Spain) isabel.cura@salud.madrid.org.

## Coordinating Committee

José María Abad-Díez (Department of Health, Social Welfare and Family, Government of Aragon), Marta Alcaraz Borrajo (Subdirectorate General of Pharmacy and Health Products), Paula Ara Bardají (Aragonese Institute of Health Sciences (IACS), IIS Aragón, Miguel Servet University Hospital, Spain), Gloria Ariza Cardiel (Research unit. Primary Health Care Management Madrid. Spain), Mercedes Aza-Pascual-Salcedo (Primary Care Department, Aragonese Health Service.), Amaya Azcoaga Lorenzo (Pintores Primary Health Care Centre, Madrid, Spain), Ana Cristina Bandrés-Liso (Primary Care Department, Aragonese Health Service.), Mercedes Clerencia-Sierra (Unit of Social and Health Assessment, Miguel Servet University Hospital, Aragonese Health Service), Nuria García-Agua (Department of Pharmacology, Faculty of Medicine, Malaga University), Luis A. Gimeno Feliu (San Pablo Primary Health Care Centre, Aragon Health Service, Zaragoza, Spain), Antonio Gimeno-Miguel (Aragonese Institute of Health Sciences (IACS), IIS Aragón, Miguel Servet University Hospital, Spain), Ana I González González (Technical Support Unit, Primary Care Management, Madrid Health Service), Virginia Hernández Santiago (Ninewells Hospital & Medical School, Dundee, UK), Francisca Leiva Fernández (Multiprofessional Teaching Unit for Family and Community Care Primary Care District Málaga-Guadarhorce. Málaga), Ana Mª López-León (Alhaurín el Grande Health Center, Malaga / Guadalhorce Sanitary District), Juan A López Rodríguez (Research unit. Primary Health Care Management Madrid. Spain), Cristina M Lozano Hernández (Research unit. Primary Health Care Management Madrid. Spain), María Isabel Márquez-Chamizo (Carranque Health Center, Malaga / Guadalhorce Sanitary District.), Alessandra Marengoni (Department of Clinical and Experimental Sciences, University of Brescia, Brescia, Italy), Javier Marta-Moreno (Department of Neurology, University Hospital

Miguel Servet, Aragonese Health Service.), Jesús Martín Fernández (Villamanta Primary Health Care Centre, Madrid, Spain), Angel Mataix SanJuan (Subdirección General de Farmacia y Productos Sanitarios), Carmina Mateos-Sancho (Ciudad Jardín Health Center, Malaga / Guadalhorce Sanitary District), Christiane Muth (Institute of General Practice, Johann Wolfgang Goethe University, Frankfurt, Germany), Victoria Pico Soler (Torrero-LaPaz Health Center, Zaragoza, Spain), Beatriz Poblador Plou (Aragonese Institute of Health Sciences (IACS), IIS Aragón, Miguel Servet University Hospital, Spain), Elena Polentinos Castro (Research unit. Primary Health Care Management Madrid. Spain), Antonio Poncel-Falcó (Primary Care Department, Aragonese Health Service.), Ricardo Rodríguez Barrientos (Research unit. Primary Health Care Management Madrid. Spain), José María Ruiz-San-Basilio (Coín Health Center, Malaga / Guadalhorce Sanitary District), Mercedes Rumayor Zarzuelo (6 Centro de Salud Pública de Coslada, Área II Subdirección de Promoción de la Salud y Prevención), Luis Sánchez Perruca (Dirección Sistemas de Información, Gerencia Asistencial de Atención Primaria, Servicio Madrileño de Salud), Teresa Sanz Cuesta (Research unit. Primary Health Care Management Madrid. Spain), Mª Eugenia Tello Bernabé (El naranjo Primary Health Care Centre, Madrid, Spain.), José María Valderas Martínez (University of Exeter Medical School, Exeter, UK. 22Department), Rubén Vázquez-Alarcón (Vera Health Center, AGS Norte de Almería).

## Clinical Investigators in Primary Healthcare Centres (PHC) MULTIPAP GROUP

**(Andalucía): PCHC Alhaurín el Grande** Javier Martín Izquierdo, Macarena Toro Sainz. **PCHC Carranque Andalucía):** Mª José Fernández Jiménez, Esperanza Mora García, José Manuel Navarro Jiménez.**PCHC Ciudad Jardín Andalucía)::** Deborah Gil Gómez, Leovigildo Ginel Mendoza, Luz Pilar de la Mota Ybancos, Jaime Sasporte Genafo.**PCHC Coín Andalucía)::** Mª José Alcaide Rodríguez, Elena Barceló Garach, Beatriz Caffarena de Arteaga, Mª Dolores Gallego Parrilla, Catalina Sánchez Morales.**PCHC Delicia Andalucía): s:** Mª del Mar Loubet Chasco, Irene Martínez Ríos, Elena Mateo Delgado.**PCHC La Roca Andalucía)::** Esther Martín Aurioles.**PCHC Limonar Andalucía)::** Sylvia Hazañas Ruiz.**PCHC Palmilla Andalucía)::** Nieves Muñoz Escalante.**PCHC Puerta Blanca Andalucía)::** Enrique Leonés Salido, Mª Antonia Máximo Torres, Mª Luisa Moya Rodríguez, Encarnación Peláez Gálvez, José Manuel Ramírez Torres, Cristóbal Trillo Fernández. **PCHC Tiro Pichón Andalucía):** Mª Dolores García Martínez Cañavate, Mª del Mar Gil Mellado, Mª Victoria Muñoz Pradilla. **PCHC Vélez Sur Andalucía):** Mª José Clavijo Peña, José Leiva Fernández, Virginia Castillo Romero.**PCHC Victoria Andalucía):** Rafael Ángel Maqueda, Gloria Aycart Valdés, Miguel Domínguez Santaella, Ana Mª Fernández Vargas, Irene García, Antonia González Rodríguez, Mª Carmen Molina Mendaño, Juana Morales Naranjo, Catalina Moreno Torres, Francisco Serrano Guerra. **Aragón: PCHC Alcorisa** (Alcorisa): Carmen Sánchez Celaya del Pozo.**PCHC Delicias Norte** (Zaragoza): José Ignacio Torrente Garrido, Concepción García Aranda, Marina Pinilla Lafuente, Mª Teresa Delgado Marroquín.**PCHC Picarral** (Zaragoza): Mª José Gracia Molina, Javier Cuartero Bernal, Mª Victoria Asín Martín, Susana García Domínguez. **PCHC Fuentes de Ebro** (Zaragoza): Carlos Bolea Gorbea.**PCHC Valdefierro** (Zaragoza): Antonio Luis Oto Negre. **PCHC Actur Norte** (Zaragoza): Eugenio Galve Royo, Mª Begoña Abadía Taira.**PCHC Alcañiz** (Alcañiz): José Fernando Tomás Gutiérrez. **PCHC Sagasta—Ruiseñores** (Zaragoza): José Porta Quintana, Valentina Martín Miguel, Esther Mateo de las Heras, Carmen Esteban Algora. **PCHC Ejea** (Ejea de los Caballeros): Mª Teresa Martín Nasarre de Letosa, Elena Gascón del Prim, Noelia Sorinas Delgado, Mª Rosario Sanjuan Cortés. **PCHC Canal Imperial—Venecia** (Zaragoza): Teodoro Corrales Sánchez. **PCHC Canal**

Imperial—**San José Sur** (Zaragoza): Eustaquio Dendarieta Lucas. **PCHC Jaca** (Jaca): Mª del Pilar Mínguez Sorio. Virginia López Cortés.**PCHC Santo Grial** (Huesca): Adolfo Cajal Marzal. **Madrid. PCHC Mendiguchía Carriche** (Leganés): Eduardo Díaz García, Juan Carlos García Álvarez, Francisca García De Blas González, Cristina Guisado Pérez, Alberto López García Franco, Mª Elisa Viñuela Benitez. **PCHC El Greco** (Getafe): Ana Ballarín González, Mª Isabel Ferrer Zapata, Esther Gómez Suarez, Fernanda Morales Ortiz, Lourdes Carolina Peláez Laguno, José Luis Quintana Gómez, Enrique Revilla Pascual. **PCHC Cuzco** (Fuenlabrada): M Ángeles Miguel Abanto.**PCHC El Soto** (Móstoles): Blanca Gutiérrez Teira. **PCHC General Ricardos** (Madrid): Francisco Ramón Abellán López, Carlos Casado Álvaro, Paulino Cubero González, Santiago Manuel Machín Hamalainen, Raquel Mateo Fernández, Mª Eloisa Rogero Blanco, Cesar Sánchez Arce.**PCHC Ibiza** (Madrid): Jorge Olmedo Galindo. **PCHC Las Américas** (Parla): Claudia López Marcos, Soledad Lorenzo Borda, Juan Carlos Moreno Fernández, Belén Muñoz Gómez, Enrique Rodríguez De Mingo. **PCHC Mª Ángeles López** (Leganés): Juan Pedro Calvo Pascual, Margarita Gómez Barroso, Beatriz López Serrano, Mª Paloma Morso Peláez, Julio Sánchez Salvador, Jeannet Dolores Sánchez Yépez, Ana Sosa Alonso. **PCHC Mª Jesús Hereza** (Leganés): Mª del Mar Álvarez Villalba. **PCHC Pavones** (Madrid): Purificación Magán Tapia. **PCHC Pedro Laín Entralgo** (Alcorcón): Mª Angelica Fajardo Alcántara, Mª Canto De Hoyos Alonso, Mª Aránzazu Murciano Antón. **PCHC Pintores** (Parla): Manuel Antonio Alonso Pérez, Ricardo De Felipe Medina, Amaya Nuria López Laguna, Eva Martínez Cid De Rivera, Iliana Serrano Flores, Mª Jesús Sousa Rodríguez. **PCHC Ramón y Cajal** (Alcorcón): Mª Soledad Núñez Isabel, Jesús Mª Redondo Sánchez, Pedro Sánchez Llanos, Lourdes Visedo Campillo.

## Author Contributions

**Conceptualization:** Cristina M. Lozano-Hernández, Amaia Calderón-Larrañaga, Isabel del Cura-González.

**Data curation:** Juan A. López-Rodríguez, Beatriz Poblador-Plou.

**Formal analysis:** Cristina M. Lozano-Hernández, Isabel del Cura-González.

**Funding acquisition:** Cristina M. Lozano-Hernández, Isabel del Cura-González.

**Investigation:** Cristina M. Lozano-Hernández, Isabel del Cura-González.

**Methodology:** Cristina M. Lozano-Hernández, Francisca Leiva-Fernández, Luis A. Gimeno-Feliu, Isabel del Cura-González.

**Project administration:** Isabel del Cura-González.

**Resources:** Cristina M. Lozano-Hernández, Isabel del Cura-González.

**Software:** Juan A. López-Rodríguez.

**Supervision:** Isabel del Cura-González.

**Validation:** Juan A. López-Rodríguez.

**Visualization:** Cristina M. Lozano-Hernández, Isabel del Cura-González.

**Writing – original draft:** Cristina M. Lozano-Hernández, Juan A. López-Rodríguez, Amaia Calderón-Larrañaga, Isabel del Cura-González.

**Writing – review & editing:** Cristina M. Lozano-Hernández, Juan A. López-Rodríguez, Francisca Leiva-Fernández, Amaia Calderón-Larrañaga, Jaime Barrio-Cortes, Luis A. Gimeno-Feliu, Isabel del Cura-González.

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
