## [Decision Letter · Decision Letter 0]

7 Feb 2020

PONE-D-20-01497

Social support, social context and nonadherence to treatment in patients with multimorbidity and polypharmacy in primary care. MULTIPAP Study.

PLOS ONE

Dear MS Lozano Hernández,

Thank you for submitting your manuscript to PLOS ONE. After careful consideration, we feel that it has merit but does not fully meet PLOS ONE’s publication criteria as it currently stands. Therefore, we invite you to submit a revised version of the manuscript that addresses the points raised during the review process.

We would appreciate receiving your revised manuscript by Mar 23 2020 11:59PM. To enhance the reproducibility of your results, we recommend that if applicable you deposit your laboratory protocols in protocols.io, where a protocol can be assigned its own identifier (DOI) such that it can be cited independently in the future. For instructions see: http://journals.plos.org/plosone/s/submission-guidelines#loc-laboratory-protocols

We look forward to receiving your revised manuscript.

Kind regards,

Delphine De Smedt

Academic Editor

PLOS ONE

Journal Requirements:

2. Please refrain from stating p values as 0.00, either report the exact value or employ the format p<0.001.

4. One of the noted authors is a group or consortium: MULTIPAP GROUP. In addition to naming the author group, please list the individual authors and affiliations within this group in the acknowledgments section of your manuscript. Please also indicate clearly a lead author for this group along with a contact email address.

Reviewers' comments:

Reviewer's Responses to Questions

**Comments to the Author**

1. Is the manuscript technically sound, and do the data support the conclusions?

Reviewer #1: Yes

Reviewer #2: Yes

2. Has the statistical analysis been performed appropriately and rigorously? 

Reviewer #1: Yes

Reviewer #2: Yes

3. Have the authors made all data underlying the findings in their manuscript fully available?

Reviewer #1: Yes

Reviewer #2: Yes

4. Is the manuscript presented in an intelligible fashion and written in standard English?

Reviewer #1: Yes

Reviewer #2: Yes

5. Review Comments to the Author

Reviewer #1: The present study addresses a very important topic of social context and nonadherence to treatment in older community dwelling patient with multimorbidity and polypharamcy.

The paper is logically structured, methodologically correctly performed and well-written.

I would recommend it for publication subject the following minor revisions.

The term elderly should be avoided throughout the manuscript and replaced by older people/patients (depending on the context).

The authors should explain somewhat more in detail why the patients with a life-expectancy less than 12 months have been excluded.

Where there any differences between adherent and nonadherent patients with regard to underlying multimorbidity and related polypharmacy?

In the discussion section, which otherwise has been fluently written and provides interesting insights I would recommend starting with the strengths of the study and the added value in comparison to existing evidence followed by limitations.

Reviewer #2: The paper aims to estimate the prevalence of nonadherence to treatment and its relationship with social support in patients with multimorbidity and polypharmacy, which is of major interest since the prevalence of multimorbidity is rising. The manuscript is well written, however, I think the paper would benefit from providing the reader with more information and further adaptations.

INTRODUCTION

- General comment: please provide line numbering.

- “Population aging has led multimorbidity..”: population aging only? And what about population growth?

- Pay attention to language: “a very common situation”

- Cut-off for multimorbidity: three or more is also recommended in the elderly.

- “It affects 81.5%...”: please clarify what you mean by “it”.

- The structure of the introduction could be improved. There are too many separate paragraphs while they can be embedded into each other in order to create more consistency. Currently, the information is fragmented. The transitions are also not clear.

- “Underutilization of necessary treatments”. Why? Please explain.

- “50% of patients with multimorbidity do not comply…”: what are the characteristics of these patients (age, disease status etc.)?

- “… for patients with cardiovascular disease.”: why do you specifically mention this disease?

- “Components of the social context are the physical environment..”: in which specific manner does the physical environment influence drug adherence? Please explain.

- Please review the use of commas in the manuscript.

- Clarify the patients within your research question, for example “in young elderly patients with multimorbidity and polypharmacy”. You could also mention this in the title of the manuscript to be transparent to the reader.

- “followed-up in primary care” � attending primary care

- “other factors of the social context”: such as? Please explain these factors in brackets.

METHODS

- Please clarify the “analytical approach”.

- Please provide more information on how multimorbidity was measured: how many diseases where measured, which diseases were involved, explain why you used a cut off value of ≥ 3 diseases to define multimorbidity, self-reported or diagnosed by a GP? Only CHRONIC diseases?

- Please provide more information about the interview during consultation. What was the methodology of the interview? Was it a structured interview? Did the GP went through the validated questionnaires with the patient? Please explain.

- Please provide more details about the Morisky-Green test. Validity? Items?

- You measured structural and functional social support. Why did you not measure the third item of social support, namely the informative item as mentioned in the introduction?

- You measured structural support based on marital status and number of cohabitants in the home. Is there a validated instrument for measuring structural support?

- Retirement status: which categories? How defined?

- CNO-11: please provide more information about this instrument.

- It is not clear how you determined socioeconomic status: is it based on educational level? Income? Occupational status? What do you mean by social class and socioeconomic level? Please clarify.

- You use income level as a surrogate for the individual's socioeconomic status? What are the cut-off values for income based on? In my opinion, this does not adequately reflect socioeconomic status. Why did you made this choice? This must be further explained in the limitations.

- EQ5D5L  EQ-5D-5L

- What are the implications of using the EQ-5D?

- Please provide more information on the EQ-5D-5L: which value set did you use to obtain the utilities? Are you interested in both the utilities and VAS scores? Please explain the difference between both scores.

- There is a difference between quality of life and health-related quality of life. A clear distinction must be made. Both are used interchangeably in this paper.

- What is the reference number of the approval of the Ethics Committee?

RESULTS + DISCUSSION

- What was the response rate?

- “errors in self-observation”: please clarify.

- Why do you think specifically these five items of social support are associated with nonadherence?

- “Cuellar-Flores et al”: provide the year of this publication in brackets

- “a representative sample of the general population”: please explain.

- Comparison with other studies, first paragraph: and what about the influence of polypharmacy and multimorbidity?

- “the experiences lived by patients in the context of multimorbidity”: what was the age class of these patients?

- “… passively cope…”: this is interesting. What could be the explanation why these patients experience a positive effect of incomprehension from their environment?

- You often mention qualitative research papers; is there also quantitative evidence?

- HRQoL: you compare the VAS score with those aged 65 years older without information on multimorbidity and polypharmacy, and the VAS scores are almost similar. This is interesting because the patients in the study are patients with multimorbidy and polypharmacy. Are there any explanations for this result?

- Implications of the study findings: can you give an practical example of how the socioeconomic context/quality of life/preferences of the patient could be taken into account when anticipating towards a more effective approach?

- “reflecting aspects linked to both the physical and social environment…”: and what about the mental component?

6. PLOS authors have the option to publish the peer review history of their article (what does this mean?). If published, this will include your full peer review and any attached files.

Reviewer #1: No

Reviewer #2: No

---

## [Author Response · Author response to Decision Letter 0]

21 May 2020

We are very grateful for your evaluation of our manuscript entitled "Social support, social context and nonadherence to treatment in patients with multimorbidity and polypharmacy in primary care. MULTIPAP Study." 

We appreciate the comments of the reviewers and editors and have made the suggested changes.

We have attached the reviewers' response.

Following the requirements of the journal, the manuscript has been reviewed with the style requirements of PLOS ONE. The p values have been modified and we have listed the individual authors and the MULTIPAP group affiliations indicating the contact email address of the main author of this group. 

We hope that these changes will contribute to improve the quality of the manuscript and the interest of potential readers.

---

## [Decision Letter · Decision Letter 1]

10 Jun 2020

Social support, social context and nonadherence to treatment in young senior patients with multimorbidity and polypharmacy followed-upin primary care. MULTIPAP Study.

PONE-D-20-01497R1

Dear Dr. Lozano Hernández,

We’re pleased to inform you that your manuscript has been judged scientifically suitable for publication and will be formally accepted for publication once it meets all outstanding technical requirements.

Kind regards,

Delphine De Smedt

Academic Editor

PLOS ONE

Additional Editor Comments (optional):

Reviewers' comments:

Reviewer's Responses to Questions

**Comments to the Author**

1. If the authors have adequately addressed your comments raised in a previous round of review and you feel that this manuscript is now acceptable for publication, you may indicate that here to bypass the “Comments to the Author” section, enter your conflict of interest statement in the “Confidential to Editor” section, and submit your "Accept" recommendation.

Reviewer #1: All comments have been addressed

Reviewer #2: All comments have been addressed

2. Is the manuscript technically sound, and do the data support the conclusions?

Reviewer #1: Yes

Reviewer #2: Yes

3. Has the statistical analysis been performed appropriately and rigorously? 

Reviewer #1: Yes

Reviewer #2: Yes

4. Have the authors made all data underlying the findings in their manuscript fully available?

Reviewer #1: Yes

Reviewer #2: Yes

5. Is the manuscript presented in an intelligible fashion and written in standard English?

Reviewer #1: Yes

Reviewer #2: Yes

6. Review Comments to the Author

Reviewer #1: The authors sufficiently addressed the reviewers' remarks and suggestions and revised the manuscript accordingly.

Consequently, I would recommend the amended version for publication.

Reviewer #2: (No Response)

7. PLOS authors have the option to publish the peer review history of their article (what does this mean?). If published, this will include your full peer review and any attached files.

Reviewer #1: No

Reviewer #2: No

---

## [Editor Report · Acceptance letter]

12 Jun 2020

PONE-D-20-01497R1 

Social support, social context and nonadherence to treatment in young senior patients with multimorbidity and polypharmacy followed-up in primary care. MULTIPAP Study.  

Dear Dr. Lozano Hernández:

I'm pleased to inform you that your manuscript has been deemed suitable for publication in PLOS ONE. Congratulations! Your manuscript is now with our production department. 

Kind regards, 

on behalf of

Dr. Delphine De Smedt 

Academic Editor

PLOS ONE